# A novel rapid detection method for a single-nucleotide substitution mutation derived from canine urothelial and prostatic carcinoma cells present in small amounts in urine sediments

**Shiro Okumura** [1]*, **Yoshiharu Ohsato** [2]

**1** Biotechnology and Food Research Institute, Fukuoka Industrial Technology Center, Kurume, Fukuoka, Japan, **2** Kahotechno Co., Ltd., Iiduka, Fukuoka, Japan

* sokumura@fitc.pref.fukuoka.jp

**Data Availability Statement:** The data presented in this study has been registered in online of DDBJ (https://www.ddbj.nig.ac.jp/). Accession numbers

## Abstract

For early detection of canine urothelial and prostatic carcinoma, we intend to develop and commercialize a simple and rapid detection method for the *BRAF* V595E mutation, a known mutation in this cancer. Detection of the single-nucleotide substitution in cancer cells contained in urine sediments is effective for early cancer diagnosis. However, urine sediment also contains many normal cells, and when there is a small relative composition of cancer cells, the mutation is difficult to detect by conventional methods other than next-generation sequencing. Our new detection method enables reliable discrimination with the same labor and cost as the PCR method. We compared the results of our new method with the results of the conventional Sanger method for 38 canine urine sediment samples, and the results of 34 samples were consistent between both methods. The remaining four results were all determined to be negative by the Sanger method and positive by our new method. For these four samples, the ratio of the mutated gene to the wild-type gene was estimated using a third-generation sequencer, and the ratio of the mutated gene was 0.1%–1.4%. We postulate that the Sanger method gave a negative result because of the low abundance of the mutated gene in these samples, proving the high sensitivity of our new method.

## Introduction

We have developed a simple and highly sensitive detection method for the canine *BRAF* V595E mutation commonly found in canine urothelial and prostatic carcinoma cells [1]. This single-base substitution mutation is usually heterozygous, so if the cells in the sample are all cancer cells, the mutation will be present at a ratio of 50%. Detection of this mutation in urine sediment is used for cancer screening, and as many normal cells are also present in urine sediment, it is very important for early diagnosis to detect mutations when the abundance of the gene mutation is low.

of the runs with MinION are below: DRA016251 (sample ID: 1-4, 1-6, 1-11, 1-20), DRA016249 (sample ID: 1-12, 1-14, 1-15, 1-17), DRA016250 (sample ID: 2-2, 2-3, 2-8, 2-9, 2-11, 2-12).

**Funding:** This study was supported by Kurume Research Park Co., Ltd. (https://www.krp.ktarn.or.jp/) from 2019 to 2021. S Okumura and Y Ohsato were jointly received the grant with no number. Provider is an organization, not an individual. The funder had no role in study design, data collection and analysis, decision to publish, or preparation of the manuscript.

**Competing interests:** The authors have declared that no competing interests exist.

In recent years, the genetic diagnosis of pets has become established as a commercial service, and Kahotechno, run by one of the authors, is engaged in such genetic diagnosis. In these commercial services, diagnosis of single-nucleotide substitutions is done with Sanger sequencing, the TaqMan method [2], single strand conformation polymorphism [3] and, in the presence of appropriate restriction enzymes, restriction fragment length polymorphism [4]. When the abundance ratio of the mutated gene in the urine sediment sample is sufficiently high, it can easily be detected by these methods, but if it is low, detection becomes challenging. Next-generation sequencing [5] have sufficient ability, but it is not suitable for the detection of only one single-nucleotide substitution because of the high cost of these highly advanced methods. And digital PCR [6] also have sufficient ability, but the disadvantage of that is the equipment is more expensive and the measurement time is longer than that of real-time PCR. There is also the "amplification refractory mutation system" [7], which is a modified PCR method, and a modified "loop-mediated isothermal amplification method" [8] for single-nucleotide polymorphisms [9]. We have found that these methods gave many false positives and inconsistent results. As a low-cost detection method for single-nucleotide substitution mutations with a low abundance ratio, oligoribonucleotide interference-PCR [10, 11] is also available. Although this is an attractive determination method, the RNA required is easily degraded and the synthesis cost is considerably higher than that of DNA. Thus, we worked on developing a low-cost, simple method to detect the *BRAF* V595E mutation.

## Materials and methods

### Oligonucleotides and Sanger sequencing

All the oligonucleotides used in our experiments (Table 1) were obtained from Eurofins Genomics Japan (Tokyo, Japan). The canine genome sequence and positional relationships of the candidate primers and blockers are summarized in Fig 1. The Sanger sequencing was also outsourced to Eurofins Genomics Japan.

We designed 14 primers and six blockers for the development of a mutated gene detection method that amplifies only the single-nucleotide mutation but not the wild-type genes. Those amplified on the upstream side of the single-nucleotide mutation position are shown above, and those amplified on the downstream side are shown below. Wild-type bases at the mutation position are shown in blue letters, and mutated bases are shown in red. Blockers were designed to be complementary to the wild-type gene, and primers were designed to be complementary to the mutated gene. Since the balance between the phase transition temperatures of primers and blockers is important in this detection method, three or more candidates with different phase transition temperatures were designed. The phase transition temperature was calculated by the method of Breslauer et al. [12] using the site Primer3 [13] with the default values except for 50 mM potassium and sodium ions, 2 mM magnesium ions, and 0.2 mM dNTPs.

### Preparation of standard samples

Three kinds of template DNA solution—mutated gene standard (Mu), wild-type gene standard (WT), and salmon sperm DNA (negative control 1; NC1)—were used in this study. To prepare Mu and WT, a 714-bp DNA amplification product with a mutation site in the center was obtained by PCR with Primer_Std_Fw and Primer_Std_Rv (Table 1), using a DNA extract obtained from canine urine sediments as the template. The amplified product was introduced into a plasmid with Mighty TA-cloning Kit (Takara Bio Inc., Shiga, Japan), and plasmids containing the mutated or wild-type gene were obtained. Mu mimics a canine genome extract at a DNA concentration of 50 ng/μL composed entirely of mutated genes. It contains $1.8 \times 10^4$ copies/μL of the TA cloning vector with the amplified mutated gene and 50 ng/μL salmon sperm (Takara Bio Inc.) solution. The number of vectors were determined on the assumption

**Table 1. Oligonucleotides used in this study.**

| Name | Size | Sequence (5′ → 3′) |
|---|---|---|
| **For selection** [a] | | |
| Primer F1 | 20 | TCCTTTACTTACTACACCTC |
| Primer F2 | 20 | &&&&&&&&&&&CACCTCAGATATTTTTCTTC |
| Primer F3 | 21 | &&&&&&&&&&&CACCTCAGATATTTTTCTTCA |
| Primer F4 | 18 | &&TTTTGGTCTAGCCACAGA |
| Primer F5 | 19 | &ATTTTGGTCTAGCCACAGA |
| Primer F6 | 20 | GATTTTGGTCTAGCCACAGA |
| Primer R1 | 17 | &&ACTCCATCGAGATTTCT |
| Primer R2 | 18 | &CACTCCATCGAGATTTCT |
| Primer R3 | 19 | CCACTCCATCGAGATTTCT |
| Primer_R4 | 19 | &TTAATTAATGGAGAAATGG |
| Primer R5 | 20 | TTTAATTAATGGAGAAATGG |
| Primer R6 | 20 | &&ATTCTTACCATCCACAAAAT |
| Primer R7 | 20 | &AATTCTTACCATCCACAAAA |
| Primer R8 | 20 | CAATTCTTACCATCCACAAA |
| Blocker BR1 | 21 | &TCGAGATTTCACTGTGGCTAG-P |
| Blocker BR2 | 23 | ATCGAGATTTCACTGTGGCTAGA-P |
| Blocker BR3 | 24 | ATCGAGATTTCACTGTGGCTAGAC-P |
| Blocker BF1 | 21 | &&CTAGCCACAGTGAAATCTCGA-P |
| Blocker BF2 | 23 | &TCTAGCCACAGTGAAATCTCGAT-P |
| Blocker BF3 | 25 | GTCTAGCCACAGTGAAATCTCGATG-P |
| **For TA cloning** | | |
| Primer_Fw_TA | 21 | CACCTCAGATATTTTTCTTCA |
| Primer_Rv_TA | 20 | CAATTCTTACCATCCACAAA |
| **For Nanopore** | | |
| BRAF_Fw_BC1 | 60 | P-ATCGCCTACCGTGAC-AAGAAAGTTGTCGGTGTCTTTGTG-CACCTCAGATATTTTTCTTCA |
| BRAF_Fw_BC2 | 60 | P-ATCGCCTACCGTGAC-TCGATTCCGTTTGTAGTCGTCTGT-CACCTCAGATATTTTTCTTCA |
| BRAF_Fw_BC3 | 60 | P-ATCGCCTACCGTGAC-GAGTCTTGTGTCCCAGTTACCAGG-CACCTCAGATATTTTTCTTCA |
| BRAF_Fw_BC4 | 60 | P-ATCGCCTACCGTGAC-TTCGGATTCTATCGTGTTTCCCTA-CACCTCAGATATTTTTCTTCA |
| BRAF_Fw_BC5 | 60 | ATCGCCTACCGTGAC-CTTGTCCAGGGTTTGTGTAACCTT-CACCTCAGATATTTTTCTTCA |
| BRAF_Fw_BC6 | 60 | ATCGCCTACCGTGAC-TTCTCGCAAAGGCAGAAAGTAGTC-CACCTCAGATATTTTTCTTCA |
| BRAF_Rv | 20 | P-CAATTCTTACCATCCACAAA |
| **For making standard samples** | | |
| Primer_Std_Fw | 20 | ATTTCAAGCCCCCAAAATCT |
| Primer_Std_Rv | 20 | CTGCAGATCGTACCTGCTGA |

[a] Oligo DNAs are displayed such that sequence positions match by inserting white character "&" within the following combinations: primer F1–F3; F4–F6; R1–R3; R4–R5; R6–F8; blocker BR1–BR3; and BF1–BF3.

that one copy of the mutated gene exists in the canine genome. WT also mimics a canine genome extract composed entirely of the wild-type gene. NC1 was a negative control, consisting of a salmon sperm gene solution at a concentration of 50 ng/μL without any canine genes. In addition, distilled water (NC2) was also used as a negative control.

## PCR conditions to select appropriate primer and blocker combinations

Screening of primers and a blocker was examined by PCR with a final concentration of 0.4 μM blocker (an oligo-DNA with phosphoric acid at the 3′ end to inhibit primer binding to the

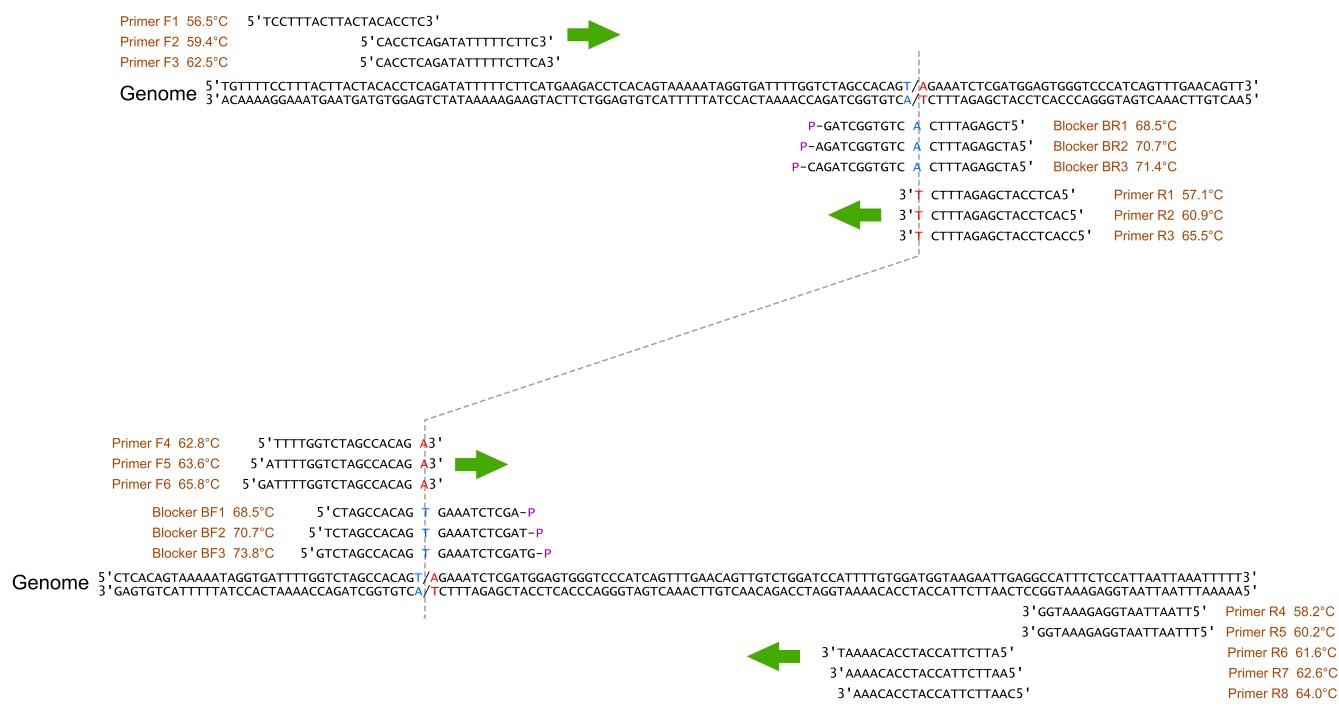

**Fig 1. Detection of the single-nucleotide mutation by PCR with a blocker.**

wild-type gene) in 10-μL reaction sets each containing 40 ng of the template DNA, 5 μL of TB Green Premix Ex Taq II (Tli RNaseH Plus) (Takara Bio Inc.), and a final concentration of 0.4 μM for each primer. The two-step PCR reaction was performed using a 30 s preheating step at 95°C, followed by 90 cycles, each consisting of denaturing at 95°C for 5 s, annealing and extension at 59°C for 30 s.

## Sensitivity test

A sample in which the standard samples Mu and WT were mixed in equal amounts was prepared to give a sample with the largest number of mutated genes, containing 9000 copies/μL of the mutated gene. Then, this solution was diluted five times in three-fold serial dilutions with the WT solution. One μL of the six sensitivity test samples prepared contained 50 ng of salmon DNA, more than 9000 copies of the plasmid with the wild-type gene, and then 9000, 3000, 1000, 333, 111, and 37 copies of the plasmid with the mutated gene, respectively. This mimics DNA extracts from a canine urine sediment with different cancer cell abundances. A sensitivity test was performed using a 1 μL sample under the same PCR conditions as in the previous section, except for the amount of template DNA and the concentration of the blocker (not only 0.4 μM but also 1.2 μM).

## First verification with test samples

One of the authors selected 22 samples from among the samples for which the presence or absence of mutations had already been determined from the chromatogram of the Sanger method so that the positive and negative samples were evenly distributed. At that time samples that were difficult to judge by the Sanger method were preferentially selected, and the rest were randomly selected with the most recent samples. They were obtained from veterinary

hospitals after obtaining consent in advance for use after anonymization. The discrimination results using our new method were compared with those with the Sanger method. Using the combination of primers F4, R8, and blocker BF1 at the 0.4 μM concentration, the other author blinded to the results of the Sanger method carried out discrimination under the same PCR conditions as in the previous section, except for a decrease to 45 thermal cycles.

## Second verification with test samples

For the second verification, detection of gene mutations of newly introduced urine sediment samples was performed almost simultaneously by the two methods, our new and the Sanger methods, unlike for the first verification. Discrimination by our new method was performed by PCR with the same primer and blocker combination as the first time. However, the polymerase was changed to QuantiTect SYBR Green PCR Kit (Qiagen, Hilden, Germany), and the PCR conditions were as follows: a 15 min preheating step at 95˚C, followed by 50 cycles each consisting of denaturing at 95˚C for 30 s, annealing and extension at 59˚C for 60 s.

## Discrimination with the third-generation sequencer

First, a 137-bp region containing a mutated position was amplified with primers BRAF_Fw_BCx (the last x is a number) and BRAF_Rv (Table 1 "For Nanopore"), using a DNA extract obtained from a canine urine sediment sample as a template. Amplifications were purified with AMPure XP (Beckman Coulter Life Sciences, IN, USA) and sequenced with the Flongle flow cell and Ligation Sequencing Kit SQK-LSK109 (Oxford Nanopore Technologies, London, UK) according to the manufacturer's instructions.

The forward primers consisted of three parts, starting from the 3′ side: an annealing part, a barcode part, and a 15-mer additional base to reliably read the barcode sequence. Barcodes were attached to identify the samples, and on the first validation, four samples were sequenced simultaneously using barcodes no. 1 to 4. If there was a phosphate modification at the 5′ end of the primer, a sequencing adapter from the supplier was ligated to the amplicon and the sequence was determined from that end. Barcodes no. 1 to 4 have a 5′-phosphate modification, as does the reverse primer, thus the sequence can be read from both sides of the amplified product. However, since sequencing from one side is sufficient for discrimination of the mutated gene, barcodes no. 5 and 6, which were additionally created, were not phosphate-modified to reduce costs. On the second validation, six samples were determined simultaneously using barcodes no. 1 to 6.

A Fastq file containing all the data from a single determination of the third-generation sequencing was loaded into Visual Studio Code (Microsoft Corporation, WA, USA), and the number of sequences in which each barcode sequence and the sequence of the wild-type gene or mutated gene existed in the same sequence was counted by the search command using the regular expression, and the gene mutation ratio was calculated for each barcode.

## Results and discussion

### Overview of our new method

Although it is theoretically possible to detect single-nucleotide substitution mutations by conventional PCR by placing one of the 3′ ends of the primers at the site of the mutation, in practice, the primers to detect the mutation often amplify the wild-type gene template. Therefore, we investigated a method that adds an oligo-DNA (hereafter referred to as a blocker), which is completely complementary to the wild-type gene and has been modified with a phosphoric acid at the 3′ end so that it does not act as a primer itself. The blocker will competitively inhibit

primer binding to the wild-type gene, helping to specifically amplify only the single-base mutation. When PCR is performed with the blocker added, the blocker preferentially binds to the wild-type gene rather than the primer for the mutated gene, so no amplification from the wild-type gene template will be expected. To effectively inhibit the binding between the wild-type gene and the primer, the single-nucleotide substitution site on the blocker was placed in the center. Since the blocker has a single-base mismatch to the mutated gene, the primer that is a full match to the mutated gene binds preferentially rather than the blocker, and the mutated gene will be amplified. Fig 1 shows the primer and blocker candidates designed for our new method. Since each phase transition temperature is predicted to be important when a blocker competitively inhibits the binding between a primer and a template, candidates with different phase transition temperatures were designed. All possible combinations of the candidates were tested for amplification against the standard wild-type or mutated gene, and a suitable blocker and primers were selected.

## Selection of appropriate primer and blocker combinations

Two standard samples, WT and Mu, and two negative controls, NC1 and NC2, were prepared to select the appropriate combinations of primers and blockers. WT and Mu contain $10^4$ copies/μL of TA cloning vector inserted with the amplified wild-type or mutated gene, respectively, in a TE buffer with 50 ng/μL salmon sperm DNA. A large amount of salmon sperm DNA was added so that the abundance ratio of the single-base substitution was the same as that of the canine genome. NC1 was a 50 ng/μL salmon sperm DNA solution without the vector, and NC2 was distilled water.

For amplification upstream from the single-base substitution site, three candidate forward primers, reverse primers, and blockers were designed (Fig 1, top), and there were 27 combinations. Similarly, there were 45 combinations for the downstream side (Fig 1, bottom). The results of nine combinations with reverse primer R8 among these 72 combinations are shown in Fig 2. Although amplification with more than 50 cycles is usually inappropriate, 90 cycles of PCR were intentionally performed to check for non-specific amplification. Fig 2A shows the Ct values for the amplification of each combination. The ideal result has Mu (red circle) located at a high position and the other three—WT (blue circle), NC1 (black diamond), and NC2 (black square)—in the lower frame (no amplification). Markers for data thought to be non-specific amplification from the melting curve are shown in faint colors. Non-specific amplification should not occur, but since it is possible to eliminate the results of non-specific amplification by the phase transition temperature of the amplicon, it is thought that even if it does occur, it will not hinder the discrimination.

When using primer F6, which has the highest phase transition temperature of F4 to F6, specific amplifications from the WT template were observed with all three blockers. Although the phase transition temperature of the F6 primer is lower than that of any blocker, amplification from the wild-type gene could not be suppressed.

With blocker BF3, which has the highest phase transition temperature of BF1 to BF3, the amplifications from the Mu template were slower (F5 and F6), or non-specific amplification was observed (F4). Not surprisingly, blockers with high phase transition temperatures have been shown to inhibit PCR amplification.

Specific amplification from a negative control was obtained with the combination of BF1 and F5. As amplification from NC1 should not occur, this is thought to have been caused by contamination.

Based on the results with primer R8, two combinations were selected—F4, R8, and BF1 and F5, R8, and BF2—and tested for sensitivity. Other combinations with the other reverse primers

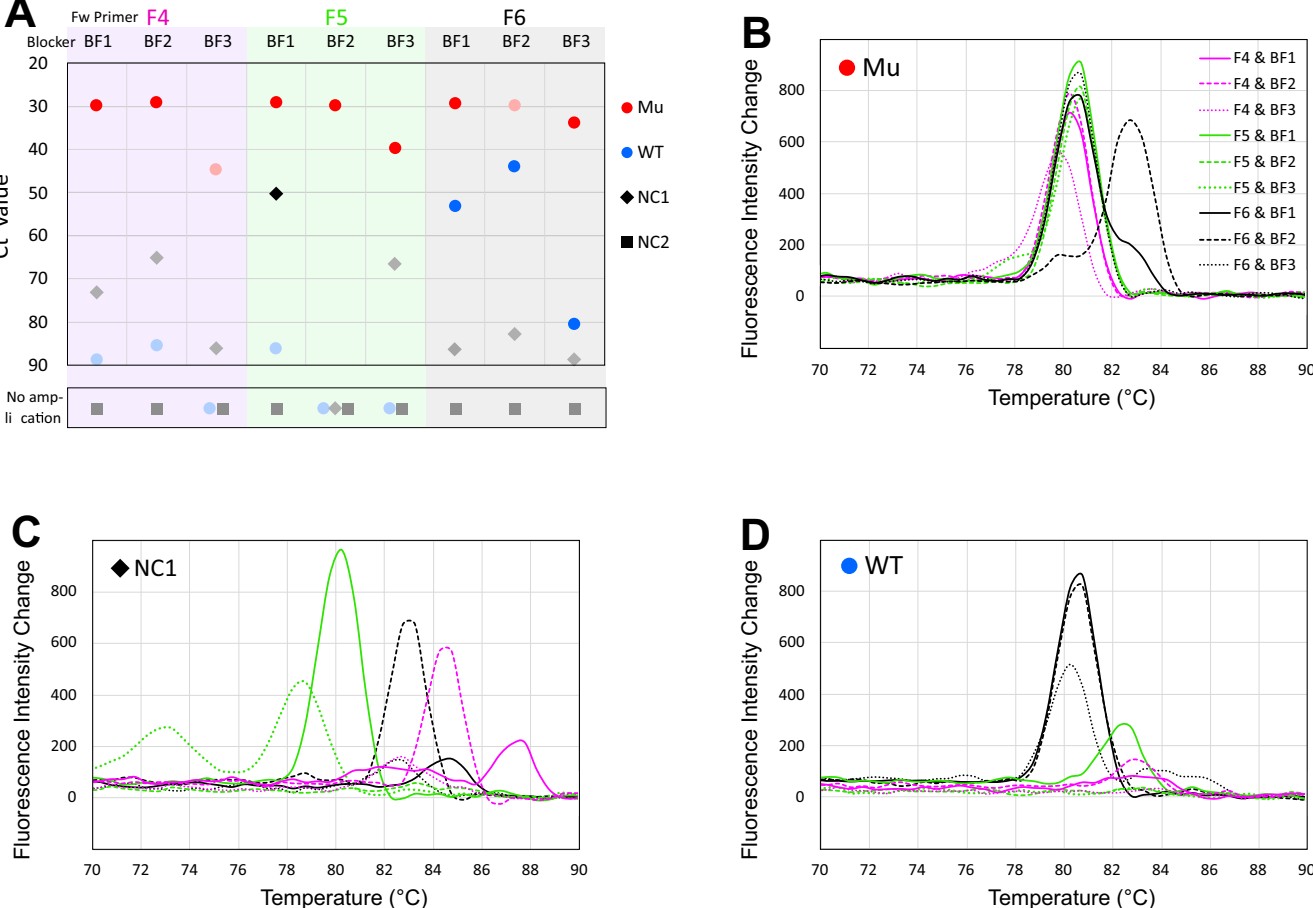

**Fig 2. Screening of primers and blockers (a case for reverse primer R8).** Ct values (**A**) and melting curves (**B-D**) of the amplified products are shown for all nine combinations with reverse primer R8. The three columns on the left of **A** show the Ct values with forward primer F4 and blockers BF1, BF2, and BF3 in order from the left. Similarly, the three central columns show those with forward primer F5, and the three right columns show those with F6. Data for Mu as the template are indicated by red circles, for WT by blue circles, for NC1 by black diamonds, and for NC2 by black squares. Markers that did not amplify within 90 cycles are placed outside below the graph area. Markers with melting curve peaks outside the range of 80˚C to 81˚C, which are judged to be non-specific amplification, are shown in faint colors. Melting curves for Mu as template are shown in **B**, for NC1 in **C**, and for WT in **D**. Melting curves with forward primer F4 are shown by the magenta line, F5 by the yellow-green line, and F6 by the black line, while melting curves with blocker BF1 are shown as a solid line, blocker BF2 as a long-dashed line, and blocker BF3 as a short-dashed line.

were screened in the same manner, and a sensitivity test of the combination of F4, R6, and BF2 was also performed.

## Sensitivity test of the three selected combinations

Sensitivity tests were performed against the three combinations selected in the previous section at two concentrations of the blocker, 0.4 μM (Fig 3A–3C) and 1.2 μM (Fig 3D–3F). Fig 3 plots the Ct value against the copy number of plasmids subcloned with the mutated gene. A sample of 9000 copies of the mutated gene was serially diluted three-fold with a WT solution that contained 9000 copies of the wild-type gene with 50 ng/μL salmon sperm DNA.

The results of the combination of F4, R6, and BF2 are shown in Fig 3A and 3D. This combination initially seemed ideal, with no unwanted specific amplification from the negative control and a good correlation between the Ct value and the copy number of the mutated gene.

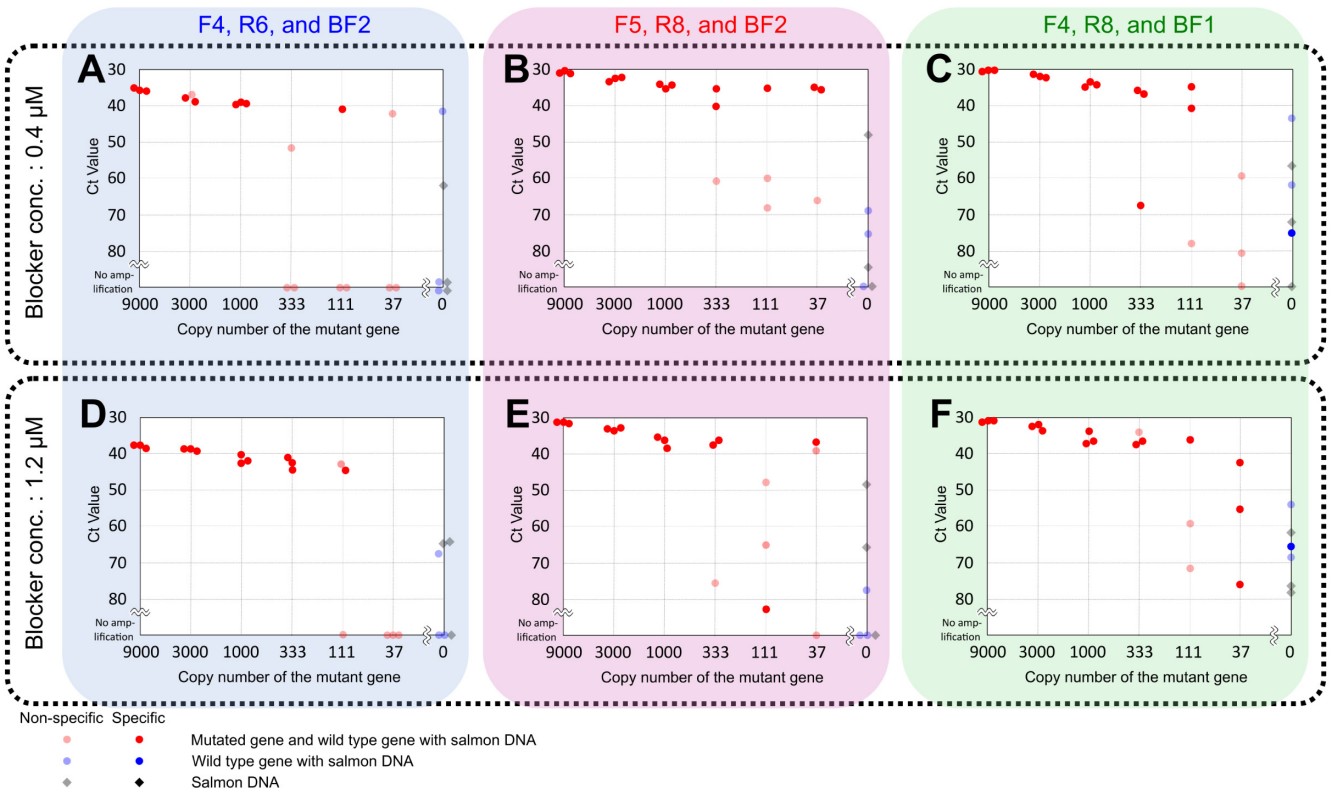

**Fig 3. Sensitivity test of the three selected detection sets.** The results of the sensitivity test for three combinations of primers and blockers—F4, R6, and BF2 (**A** and **D**); F5, R8, and BF2 (**B** and **E**); and F4, R8, and BF1 (**C** and **F**)—are shown. Two concentrations of the blocker were tested, and the results for 0.4 μM are shown in the upper row (**A**-**C**) and for 1.2 μM in the lower row (**D**-**F**). The copy numbers of the mutated gene in the test solutions were 9000, 3000, 1000, 333, 111, and 37 in order from the left, and three samples of each copy number were tested. Markers that overlap and are difficult to see are displayed by shifting their positions to the left or right horizontally. Each sample contained 50 ng of salmon DNA and at least $9 \times 10^3$ copies of plasmids containing the wild-type gene. Three samples each of WT and NC1 were tested as the negative controls, and the results are shown on the right line of the graph area. Data that did not amplify within 90 cycles are placed on the underline of the graph area, and markers with melting curve peaks out of the range between 80°C and 81°C are shown in a faint color as they were judged to be non-specific amplification.

However, the amplification rate of Mu was slower than the other combinations, requiring more than 35 cycles even for the highest copy number.

The results of the combination F5, R8, and BF2 (Fig 3B and 3E) seemed to be practical. However, the absence of a reduction in the amplification rate in low copy number samples of the mutated gene led us to suspect unwanted specific amplification from the wild-type gene, because all positive samples contained 9000 copies of the wild-type gene. Sequencing of the amplified product should have been performed for confirmation, but this was not done because throughout the study the lid of the amplified product was not opened, to prevent contamination.

Not limited to this result, the sequence data of the amplicon was considered extremely useful for confirming the results and investigating the cause of non-specific amplification. However, since the amplicons in this study are very short (81 to 114 bp) and are not easily damaged in the natural environment, thus they easily cause contamination once they are released into the air. Therefore, we decided to incinerate them without opening the lid of the tube.

The results of the combination F4, R8, and BF1 are shown in Fig 3C and 3F. In this combination, the Ct value was about 30 and the amplification was rapid where the number of mutated genes was the largest, and the correlation between the copy number and the Ct value

was good except when the number of mutated genes was the smallest. Because the results were almost the same between Fig 3C and 3F, we decided to perform verification with a blocker concentration of 0.4 μM (Fig 3C) in consideration of the cost of the procedure. In this combination, one unwanted specific amplification from the negative control was observed; however, the Ct value was about 75 and so this was judged to not be a problem.

## First verification with test samples

In dogs with early symptoms of urothelial and prostatic carcinoma, lesions are visualized by ultrasound and morphological evaluation of cell tissue collected from suspected lesions is performed. At that time, BRAF gene mutation testing is often performed as an aid to make the diagnosis more definitive. Although it has recently been reported that the presence of BRAF mutations does not necessarily correlate with the presence of clinical cancer [14], the detection of the mutation is still practiced. The urine sediments of dogs judged by a veterinarian to require discrimination of the gene mutation were subjected to examination with the combination of primers and blockers selected in the previous section (F4, R8, and BF1). These results were compared with the discrimination by chromatograms using the Sanger method (Table 2). We selected 22 samples for verification, of which 13 were determined as positive and nine negative by the Sanger method, so that the numbers of positive and negative samples were similar. For these samples, our new method determined 16 samples as positive and six

**Table 2. Results of detecting the gene mutation by the conventional method and our new method (first verification).**

| Sample ID | Sanger method | Our new method | Match or not | Ratio of gene mutation (%) [a] |
|---|---|---|---|---|
| 1–1 | + | + | Match | n.t. |
| 1–2 | + | + | Match | n.t. |
| 1–3 | + | + | Match | n.t. |
| **1–4** | **–** | **+** | **not** | **0.1** |
| 1–5 | + | + | Match | n.t. |
| **1–6** | **–** | **+** | **not** | **0.4** |
| 1–7 | + | + | Match | n.t. |
| 1–8 | + | + | Match | n.t. |
| 1–9 | + | + | Match | n.t. |
| 1–10 | + | + | Match | n.t. |
| **1–11** | **–** | **–** | **Match** | **0.0** |
| **1–12** | **–** | **–** | **Match** | **0.0** |
| 1–13 | + | + | Match | n.t. |
| **1–14** | **–** | **–** | **Match** | **0.0** |
| **1–15** | **+** | **+** | **Match** | **14.3** |
| 1–16 | – | – | Match | n.t. |
| **1–17** | **+** | **+** | **Match** | **49.6** |
| 1–18 | + | + | Match | n.t. |
| 1–19 | – | – | Match | n.t. |
| **1–20** | **–** | **+** | **not** | **1.2** |
| 1–21 | + | + | Match | n.t. |
| 1–22 | – | – | Match | n.t. |

Data for which the results of the Sanger and the new method did not match are shown in black background and white letters. Data with consistent results, measured by the third-generation sequencer for the ratio of gene mutations, are shown in gray background and bold letters.

[a] "n.t." means "not tested".

samples as negative. There were three samples with different results, all of which had been discriminated as negative by the Sanger method and positive by the new method (Table 2; 1–4, 1–6, and 1–20).

We then subcloned these three samples and ten clones for each sample were selected and determined by the Sanger method whether it is the wild-type or the mutated gene. The results showed that all clones of all samples were the wild-type gene. If a mutated gene was present, it was expected that the abundance ratio would be low, and so further subcloning was stopped and verification by next-generation sequencing was performed.

The primers shown in Table 1 "For Nanopore" were used to amplify the single-nucleotide substitution portion, and the amplicons were sequenced with a third-generation sequencer, MinION (Nanopore Technologies). MinION can determine whether each amplified product is a mutated or wild-type gene [15]. Therefore, since the abundance of gene mutations in the amplified product is considered to reflect that of the template, we estimated the gene mutation rate of the template from the MinION sequencing results. Eight samples were sequenced, including the three samples with inconsistent judgment results, and the abundance ratio of the mutated genes was calculated from 1000 or more results for each (Table 2, fifth column). As a result, the ratios of the mutated genes were 0.1%, 0.4%, and 1.2% in the three discordant samples, showing that our novel identification method can discriminate mutated genes even at around 1%.

All three samples that had shown concordant negative results (1–11, 1–12, and 1–14) had a mutation rate of 0.0% to two significant figures and were also negative on third-generation sequencing. The two samples that showed concordant positive results (1–15 and 1–17) had mutation rates of 14.3% and 49.6%, which were also positive on the third-generation sequencer. Since these five results agreed with each other on the use of all three methods, we were able to show that the determination of single-base substitutions in the amplicons using the third-generation sequencer also has a certain degree of reliability.

## Second verification with test samples

Validation similar to the previous section was repeated by changing the operator, thermal cycler, and polymerase when performing our new method. The PCR conditions were modified slightly to suit the polymerase used. Unlike the first time, mutation was determined almost simultaneously in undetermined urine sediment samples by both the Sanger and our new method.

Of the 16 samples, three were positive and 13 were negative by the Sanger method, and our new method gave four positives and 12 negatives. There was one sample with different discrimination results, which was negative by the Sanger method and positive by the new method (2–11 in Table 3). In this verification, subcloning was not performed, and verification by third-generation sequencing was performed for the one discordant, three positively matched, and two negatively matched samples. The result showed that the one discordant sample (2–11) contained mutated gene amplification products at a rate of 1.4%. It is thought that the urine sediments contained cells with the mutated gene, albeit in small numbers, thus demonstrating that our new method can detect a mutated gene with high sensitivity. As in the first verification, the results using the third-generation sequencer were also consistent for the five samples (2–2, 2–3, 2–8, 2–9, and 2–12) for which the results of the two methods were in agreement.

Combined with the results of the first round, the percentage of mutated gene amplification products in specimens determined to be positive by the Sanger method ranged from 14.3% to 49.6%, and the lower limit of detection by this method was estimated to be around 15%. Our new method, however, detected positives present at percentages of 0.1% to 1.4% in samples

**Table 3. Results of detecting the gene mutation by the conventional method and our new method (second verification).**

| Sample ID | Sanger method | Our new method | Match or not | Ratio of gene mutation (%) [a] |
|---|---|---|---|---|
| 2–1 | – | – | Match | n.t. |
| **2–2** | **–** | **–** | **Match** | **0.0** |
| **2–3** | **+** | **+** | **Match** | **43.8** |
| 2–4 | – | – | Match | n.t. |
| 2–5 | – | – | Match | n.t. |
| 2–6 | – | – | Match | n.t. |
| 2–7 | – | – | Match | n.t. |
| **2–8** | **+** | **+** | **Match** | **18.9** |
| **2–9** | **+** | **+** | **Match** | **37.6** |
| 2–10 | – | – | Match | n.t. |
| **2–11** | **–** | **+** | **not** | **1.4** |
| **2–12** | **–** | **–** | **Match** | **0.0** |
| 2–13 | – | – | Match | n.t. |
| 2–14 | – | – | Match | n.t. |
| 2–15 | – | – | Match | n.t. |
| 2–16 | – | – | Match | n.t. |

Data for which the results of the Sanger and the new method did not match are shown in black background and white letters. Data with consistent results, measured by the third-generation sequencer for the ratio of gene mutations, are shown in gray background and bold letters.

[a] "n.t." means "not tested".

that were negative by the Sanger method, demonstrating that it is much more sensitive than the Sanger method. In addition, there were no cases in which a specimen with a mutated gene abundance rate of 0.0% was discriminated to be positive, indicating that our new method is less prone to false positives.

## Conclusions

Canine urothelial and prostatic carcinoma is often found in an advanced state, and an early diagnosis method is required that is less economically and physically burdensome than current methods. Urine sediment generally contains many normal cells as well as cancer cells, so it is desirable to be able to detect the mutation even when the abundance ratio of the mutated gene is low. The TaqMan and Sanger methods are unable to detect the mutation when the abundance is low, and some of the modified PCR methods mentioned in the introduction are often not practical as they give many false positive results.

In this study, there were four samples in which the mutation was detected by the new method, and not by Sanger sequencing. It shows that our new method has high sensitivity. Nevertheless, our method did not falsely identify as "positive" for samples with mutation abundances below 0.0% on the third-generation sequencer. In terms of the time required for measurement, our method is short at less than one hour. This is an advantage compared to, for example, about 4 hours for the digital PCR. In addition, it is almost the same as the PCR method except for the addition of a blocker, and it can be performed easily with only a slight increase in cost. Moreover, since the mutation status can be determined using urine sediment, the physical burden on the subject animal is extremely low. We sincerely hope that this method will become widespread and will be incorporated into regular health checkups for dogs, and that in future, canine urothelial and prostatic carcinoma will be detected early worldwide.

## Acknowledgments

We would like to thank my colleagues, Dr. Hideki Katayama and Dr. Rieko Kuroda of the Biotechnology and Food Research Institute, for their useful advice on how to proceed and the framework of the research.

## Author Contributions

**Conceptualization:** Shiro Okumura.

**Formal analysis:** Shiro Okumura.

**Funding acquisition:** Shiro Okumura, Yoshiharu Ohsato.

**Investigation:** Shiro Okumura, Yoshiharu Ohsato.

**Methodology:** Shiro Okumura, Yoshiharu Ohsato.

**Project administration:** Yoshiharu Ohsato.

**Resources:** Yoshiharu Ohsato.

**Writing – original draft:** Shiro Okumura.

**Writing – review & editing:** Yoshiharu Ohsato.

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
