## [Decision Letter · Decision Letter 0]

6 Jul 2023

PONE-D-23-14092A novel rapid detection method for a single-nucleotide substitution mutation derived from canine urothelial and prostatic carcinoma cells present in small amounts in urine sedimentsPLOS ONE

Dear Dr. Okumura,

Thank you for submitting your manuscript to PLOS ONE. After careful consideration, we feel that it has merit but does not fully meet PLOS ONE’s publication criteria as it currently stands. Therefore, we invite you to submit a revised version of the manuscript that addresses the points raised during the review process.

We look forward to receiving your revised manuscript.

Kind regards,

Ruslan Kalendar

Academic Editor

PLOS ONE

Journal Requirements:

2.  In your manuscript, you appear to indicate that urine sediments of dogs suspected of having urothelial and prostatic carcinoma were obtained and used in your study. However, there is no description of where the urine samples were obtained and permission. Please clarify. 

Reviewers' comments:

Reviewer's Responses to Questions

**Comments to the Author**

1. Is the manuscript technically sound, and do the data support the conclusions?

Reviewer #1: Partly

Reviewer #2: Yes

2. Has the statistical analysis been performed appropriately and rigorously? 

Reviewer #1: I Don't Know

Reviewer #2: Yes

3. Have the authors made all data underlying the findings in their manuscript fully available?

Reviewer #1: Yes

Reviewer #2: Yes

4. Is the manuscript presented in an intelligible fashion and written in standard English?

Reviewer #1: Yes

Reviewer #2: Yes

5. Review Comments to the Author

Reviewer #1:

The authors present interesting work aimed at developing an inexpensive method to detect the BRAFV595E mutation in the urine of dogs. They point out the shortcomings of other assays and the expense of other assays.

The authors clearly describe their new approach, the experiments performed, and the results. Their strategy for the assay is intriguing.

The finding that there were samples that were negative for the mutation by Sanger sequencing and positive for the mutation with the new assay were not surprising, and the authors presented followup work to explain the enhanced sensitivity of their new assay. Defining the specific percentage of alleles with the mutation which are required for a positive test by either method, however, requires further study.

One of the main concerns about the research is that the results were not compared to ddPCR, which is the most widely available commercial assay to detect the BRAF mutations in dog urine. Although the ddPCR test is expensive through commercial labs, it can actually be performed for a fraction of that cost. The commercial labs overcharge for the assay. Based on the sentence starting on line 385, is the reader to conclude that the new assay is more expensive than the ddPCR assay?

On line 134, please explain how the samples were randomly selected? How many samples were available? How ere the 22 samples selected?

Could the authors explain the sentence starting on line 297 more clearly. In what instances did a veterinarian require discrimination of the gene mutation? Was it for targeted therapy, or was it related to a diagnosis? It is becoming more obvious that the presence of the mutation does not always correlate with the presence of clinical cancer. This is being recognized by multiple veterinary specialty centers, and the poor predictive value was confirmed by a study published in 2022. https://pubmed.ncbi.nlm.nih.gov/36439482/

The sentence starting on line 43 is not necessarily correct. The percentage of cells in a tumor mass that harbor the BRAFV595E mutation is thought to vary considerably from tumor to tumor. And the ratio of tumor cells to normal urothelial cells in the urine also varies from dog to dog. If there are referenced studies to back up this sentence, please add those references. Otherwise, it would be best to remove this sentence.

This same comment applies to the sentences starting on line 377.

Can the authors please clarify the sentence starting on line 384. Did the third-generation sequencing match with the results of the new assay for the 3 samples that were positive by the new assay and negative via Sanger sequencing? If the authors are confident in that result, then it’s not necessarily a “false positive”, rather it shows higher sensitivity with the new assay. It would be best to simply state that there were 3 samples in which the mutation was detected by the new method, and not by Sanger sequencing.

Reviewer #2: 

The author has successfully developed a straightforward, rapid, and highly efficient technique for the detection of single nucleotide mutation, exhibiting high sensitivity and economic feasibility. Notably, this method investigates a superiorly stable DNA blocker in comparison to the ORNi-PCR. The experimental design exhibits strong logic and coherence, while the validation of the method remains robust and satisfactory. However, I would like to propose two recommendations that might further refine this paper.

1. Figure 3 delineates data obtained from examining three primer combinations and two blocker concentrations. To better augment the legibility and understanding of this figure, would the author consider incorporating these specific annotations directly into the figure? Such an addition would simplify the interpretation of the data for the readers. Furthermore, it would be beneficial if the meaning of each dot (presumably representing an individual test) could be explicitly mentioned within the figure illustration.

2. In the section entitled "Sensitivity Test", has the author considered to sequencing the resultant products? This could potentially provide an unambiguous manner to demonstrate the efficiencies of the three primer combinations.

I look forward to hearing your thoughts on these suggestions and I appreciate your time and consideration in addressing these recommendations.

6. PLOS authors have the option to publish the peer review history of their article (what does this mean?). If published, this will include your full peer review and any attached files.

Reviewer #1: No

Reviewer #2: **Yes: **Yin Tang

While revising your submission, please upload your figure files to the Preflight Analysis and Conversion Engine (PACE) digital diagnostic tool, https://pacev2.apexcovantage.com/. PACE helps ensure that figures meet PLOS requirements. To use PACE, you must first register as a user. Registration is free. Then, login and navigate to the UPLOAD tab, where you will find detailed instructions on how to use the tool. If you encounter any issues or have any questions when using PACE, please email PLOS at figures@plos.org. Please note that Supporting Information files do not need this step.<quillbot-extension-portal></quillbot-extension-portal>

---

## [Author Response · Author response to Decision Letter 0]

3 Aug 2023

First, we are grateful to the editor and reviewers for the meticulous work in the review of this manuscript. We feel greatly appreciated that the reviewer made so many advises for us. As indicated in the responses that follow, we have taken all these comments and suggestions into account in the revised version of our manuscript. The suggestions were of great help for us to improve the quality of the manuscript. Thank you again and we are looking forward to your reply.

Comments from the Reviewer #1

Q1

One of the main concerns about the research is that the results were not compared to ddPCR, which is the most widely available commercial assay to detect the BRAF mutations in dog urine. Although the ddPCR test is expensive through commercial labs, it can actually be performed for a fraction of that cost. The commercial labs overcharge for the assay. Based on the sentence starting on line 385, is the reader to conclude that the new assay is more expensive than the ddPCR assay?

A1

There is little doubt that digital PCR can provide reliable and accurate determinations for the BRAF mutation. In addition, it has economical running costs. We thank you for pointing out that it is the highest priority comparison target for our new method in that respect. However, digital PCR equipment is currently expensive and depreciation costs, which may be the reason why it is expensive in commercial laboratories for use of that. Although it depends on the number of samples, the measurement time of the digital PCR is about 4 hours after reagent preparation, and our method is superior in this respect as well. We have added these points to the "Introduction (Line 56, Revised Manuscript with Track Changes)" and "Conclusion (Line 408)".

Q2

On line 134, please explain how the samples were randomly selected? How many samples were available? How ere the 22 samples selected?

A2

For the first verification, Ohsato selected samples, and Okumura conducted a judgment experiment using the new method without hearing the judgment results of the conventional method. First, Ohsato preferentially selected samples that were difficult to judge by the conventional method. Subsequently, the rest were selected to have a similar number of positives and negatives, starting with the most recent sample. Thank you for pointing out the inaccurate description. We have revised the "First verification with test samples (Line 136)" section of "Materials and methods" to clarify the actual selection criteria.

Q3

Could the authors explain the sentence starting on line 297 more clearly. In what instances did a veterinarian require discrimination of the gene mutation? Was it for targeted therapy, or was it related to a diagnosis? It is becoming more obvious that the presence of the mutation does not always correlate with the presence of clinical cancer. This is being recognized by multiple veterinary specialty centers, and the poor predictive value was confirmed by a study published in 2022. https://pubmed.ncbi.nlm.nih.gov/36439482/

A3

We were not aware of the article you inquired about. We thank you for pointing out the lack of our research. In fact, there is still a wide demand for determination of the mutation in BRAF to aid diagnosis, and for this reason we are conducting confirmatory tests on samples brought in from veterinarians. We have added the necessary description below line 311 and added the paper in reference 14.

Q4

The sentence starting on line 43 is not necessarily correct. The percentage of cells in a tumor mass that harbor the BRAFV595E mutation is thought to vary considerably from tumor to tumor. And the ratio of tumor cells to normal urothelial cells in the urine also varies from dog to dog. If there are referenced studies to back up this sentence, please add those references. Otherwise, it would be best to remove this sentence. 

This same comment applies to the sentences starting on line 377.

A4

We thank you for pointing out an inaccurate description. We have deleted the relevant part (Line43 and 396).

Q5

Can the authors please clarify the sentence starting on line 384. Did the third-generation sequencing match with the results of the new assay for the 3 samples that were positive by the new assay and negative via Sanger sequencing? If the authors are confident in that result, then it’s not necessarily a “false positive”, rather it shows higher sensitivity with the new assay. It would be best to simply state that there were 3 samples in which the mutation was detected by the new method, and not by Sanger sequencing.

A5

We thank you for the advice. We revised the description according to your advice (Line 404).

Comments from the Reviewer #2

Q1. Figure 3 delineates data obtained from examining three primer combinations and two blocker concentrations. To better augment the legibility and understanding of this figure, would the author consider incorporating these specific annotations directly into the figure? Such an addition would simplify the interpretation of the data for the readers. Furthermore, it would be beneficial if the meaning of each dot (presumably representing an individual test) could be explicitly mentioned within the figure illustration.

A1

We thank you for the advice. We modified Fig.3 according to your advice. In addition, we have made changes to Fig3 to make it easier to see.

Q2. In the section entitled "Sensitivity Test", has the author considered to sequencing the resultant products? This could potentially provide an unambiguous manner to demonstrate the efficiencies of the three primer combinations.

A2

We thank you for your useful advice. We also have wanted to sequence the resultant products. However, since the amplicons in this measurement are very short (81 to 114 bp) and are not easily damaged in the natural environment, they easily cause contamination once they are released into the environment. Therefore, we decided not to analyze the resultant products and incinerate them without opening the lid of the tube. We had written very briefly why we didn't analyze the resultant products from line 284 for a particular primer and blocker set combination, and we have added a new paragraph (Line 295, Revised Manuscript with Track Changes) according to your advice.

That's all.

---

## [Editor Report · Decision Letter 1]

6 Sep 2023

A novel rapid detection method for a single-nucleotide substitution mutation derived from canine urothelial and prostatic carcinoma cells present in small amounts in urine sediments

PONE-D-23-14092R1

Dear Dr. Okumura,

We’re pleased to inform you that your manuscript has been judged scientifically suitable for publication and will be formally accepted for publication once it meets all outstanding technical requirements.

Kind regards,

Ruslan Kalendar

Academic Editor

PLOS ONE

---

## [Editor Report · Acceptance letter]

12 Sep 2023

PONE-D-23-14092R1 

A novel rapid detection method for a single-nucleotide substitution mutation derived from canine urothelial and prostatic carcinoma cells present in small amounts in urine sediments 

Dear Dr. Okumura:

I'm pleased to inform you that your manuscript has been deemed suitable for publication in PLOS ONE. Congratulations! Your manuscript is now with our production department. 

Kind regards, 

on behalf of

Professor Ruslan Kalendar 

Academic Editor

PLOS ONE